# An Intrusion Detection System over the IoT Data Streams Using eXplainable Artificial Intelligence (XAI)

**DOI:** 10.3390/s25030847

**Published:** 2025-01-30

**Authors:** Adel Alabbadi, Fuad Bajaber

**Affiliations:** Faculty of Computing and Information Technology, King Abdulaziz University, Jeddah 21589, Saudi Arabia; fbajaber@kau.edu

**Keywords:** networks, IDS, machine learning, deep learning, CNN, neural networks, LIME, SHAP

## Abstract

The rise in intrusions on network and IoT systems has led to the development of artificial intelligence (AI) methodologies in intrusion detection systems (IDSs). However, traditional AI or machine learning (ML) methods can compromise accuracy due to the vast, diverse, and dynamic nature of the data generated. Moreover, many of these methods lack transparency, making it challenging for security professionals to make predictions. To address these challenges, this paper presents a novel IDS architecture that uses deep learning (DL)-based methodology along with eXplainable AI (XAI) techniques to create explainable models in network intrusion detection systems, empowering security analysts to use these models effectively. DL models are needed to train enormous amounts of data and produce promising results. Three different DL models, i.e., customized 1-D convolutional neural networks (1-D CNNs), deep neural networks (DNNs), and pre-trained model TabNet, are proposed. The experiments are performed on seven different datasets of TON_IOT. The CNN model for the network dataset achieves an impressive accuracy of 99.24%. Meanwhile, for the six different IoT datasets, in most of the datasets, the CNN and DNN achieve 100% accuracy, further validating the effectiveness of the proposed models. In all the datasets, the least-performing model is TabNet. Implementing the proposed method in real time requires an explanation of the predictions generated. Thus, the XAI methods are implemented to understand the essential features responsible for predicting the particular class.

## 1. Introduction

Intrusion detection (ID) is the process of scanning a computer system or network for unexpected behavior, such as unauthorized access, fraud, or asset tampering [1]. The purpose of IDs is to detect these acts as they occur and then take appropriate safeguards to prevent further data loss or theft. In order to identify suspicious activity that may indicate a continuing security risk, an IDS is designed to analyze system and network traffic [2]. These networks use various techniques like signature-based identification, anomaly-based identification, or behavior-based detection to recognize potential dangers; they can be host-based or network-based. So that the attack can be controlled or prevented, the IDS may inform security personnel or use automatic reaction mechanisms (such as firewalls) when an incursion is identified. To better detect and react to security issues, identification is an essential part of an all-encompassing security strategy [3].

The way in which individuals live and work has been transformed by the integration of electronic devices and the internet into all facets of life. New possibilities for telecommuting, online learning or working, and constant connection have emerged as a result of this integration. Nevertheless, security issues and data breaches might result from technology’s convenience. This includes staying vigilant against phishing scams, using strong passwords, and keeping updated software. This includes remaining aware of current security dangers and being sensitive to signs of suspicious activity [4]. Researchers and corporations alike are investing resources in developing smart solutions to the problem of IoT assaults. The objective is to improve the safety of IoT networks using approaches similar to the previous intrusion detection systems [5].

Consequently, new security measures are necessary to protect the IoT from various attacks and scanning attacks. When it comes to protecting IoT networks from multiple attacks, IDSs show significant promise. To increase the detection rate of intrusions related to the IoT, intelligent IDSs have been built using machine learning (ML) techniques. In reality, ML-based IDSs are meant to learn the specific features of each IoT attack, allowing them to be foreseen and detected quickly and correctly.

Staff members such as cybersecurity professionals or executive staff should take action to deal with an attack as soon as it is detected. On the other side, some of the recent IDSs use DL that primarily uses DNN models, which can train the data and provide the predictions without explaining how these predictions are generated; in other words, it is difficult to understand how these models work, especially for users who are not data scientists. As a result, these models are delivered and used in a black-box environment, and users only see the results of these models’ findings, with no context or analysis of their reasoning. Users are unable to understand and depend on decision-making processes created by DL models, nor can they optimize their decision-making processes based on DL model results [3].

The major limitation of the IDS research is the use of black-box models. These black-box models can make very accurate predictions, but they do not give any reasons for their predictions. This is because it is difficult to isolate the data points that impact their decision-making due to their multi-level and nonlinear structure [6]. User trust is weakened by the inability to trace results back to the original data and the lack of understanding of the workings of AI models. The black-box nature of these models creates issues in a variety of domains where AI or its components are used. When alarms from an intrusion detection system are not explained, task analysis makes decision-making more difficult [7]. In response to these restrictions, a new AI paradigm known as eXplainable Artificial Intelligence (XAI) has arisen, providing a set of tools for understanding and comprehending DL model predictions [8]. XAI enables cybersecurity experts to better understand the judgments made by DL-based IDSs. This simplifies decision-making by allowing specialists to trust and modify such models.

Thus, this paper applies XAI methods to develop transparent network intrusion detection systems by implementing two different XAI techniques. A novel IDS framework based on transfer learning and the DL approach has been developed. Extensive datasets are utilized to perform experiments and prove that the framework is efficient enough and acquires high performance on seven different datasets. Consequently, the proposed framework will extend the utilization of XAI in network intrusion detection systems, facilitating greater transparency and improvement in this study domain and identifying the most influencing features in the datasets that are responsible for predictions.

The summary of the contributions of this research is given below.

This research proposes a novel transfer learning and deep learning-based framework to create customized models. The pre-trained TabNet model is utilized, and two customized 1D-CNN and DNN are developed;The proposed framework is evaluated on seven different open-source datasets to prove its effectiveness on all types and sizes of datasets. The dataset utilized is an open-source dataset, TON-IoT, which consists of one network and six IoT datasets;The ablation study is performed to fine-tune the customized and transfer learning models to achieve the highest performance on the datasets. The models are trained and verified with different hyperparameters to choose the most optimized ones;To introduce transparency in the predicted results and understand the importance of features, XAI techniques like Local Interpretable Model-agnostic Explanations (LIME) and SHAPley Additive exPlanations (SHAP) are implemented. The LIME and SHAP results are compared to verify the efficiency of the XAI techniques and understand the essential features for prediction for these particular models.

## 2. Literature Review

Various researchers have used ML and DL to develop the IDS. Musleh et al. [9] presented a study on ML-based IDSs in IoT, taking into account several ML models and feature extraction strategies. The transfer learning models, like VGG-16 and DenseNet, were assessed in this study. In addition, a number of machine learning methods were evaluated, taking into account all of the investigated feature extraction approaches, such as random forest (RF), k-nearest neighbors (KNN), support vector machines (SVMs), and other stacked models. Using the IEEE Dataport dataset, a thorough assessment of all integrated models was presented in the paper. The best accuracy of 98.3% was obtained when stacking, and VGG-16 was used together, according to the results.

Similarly, Awajan [10] employed a four-layer deep, fully linked model for intrusion detection on IoT data. To simplify implementation, the suggested system was designed to be independent of communication protocols with the dataset that has been generated with 25,000 occurrences. The dataset’s instance ratios for classes, i.e., Regular and Malicious, are 82:18. With an average accuracy of 93.74%, it can identify black, Distributed Denial of Service, Opportunistic Service, Sinkhole, and Workhole attacks with an accuracy of 93.21%. The limitation of the research is that the model developed is not generic but system-specific, detecting only five specific intrusions. Roshan and Zafar [11] also used ensemble adaptive online ML to identify and categorize cyber intrusions in data collected from network traffic. The suggested EnsAdp_CIDS method adjusts its parameters dynamically while learning each instance individually. The highest accuracy achieved was 99.85% on CIC-MalMem-2022.

Similar to the above approach, Altulaihan et al. [12] used four supervised classifier algorithms for the detection of DDoS attacks. Furthermore, two distinct feature selection algorithms, i.e., the Genetic Algorithm (GA) and the Correlation-based Feature Selection (CFS) algorithm, were employed and evaluated. To further enhance the model’s ability to identify suspicious behavior in IoT networks, it was trained using the IoTID20 dataset, which is among the most current in this area. Both decision tree and random forest (RF) reached 99.936% and 99.9482% accuracy without feature selection, respectively; both obtained 100% accuracy with GA feature selection and 99.9859% accuracy in the correlation coefficient. Saiyed and Al-Anbagi [13] also presented a revolutionary lightweight approach for detecting DDoS attacks in IoT networks. The recommended system consists of three phases and uses edge-based technologies. The initial step is creating and pre-processing an HL-IoT dataset and then developing a new, lightweight Genetic algorithm and statistical parameter method for optimizing feature selection. Finally, the system trains three tree-based ML models, i.e., RF, Extra Tree (ET), and AdaBoost—and other ML models using the private dataset and the publicly accessible ToN-IoT dataset. The highest accuracy of 96% and 94% was achieved by the RF model on private and ToN-IoT datasets, respectively.

Mishra et al. [14] presented a weighted stacked ensemble model that combines bidirectional long short-term memory networks (Bi-LSTMs) with deep convolutional generative adversarial networks (GANs) and experimented on four recent IoT datasets. An L2 regularization strategy was utilized to improve the efficiency of the model. The highest accuracy achieved was 99.99% for the BOT-IoT dataset. Yadav et al. [15] also used the Recurrent Neural Network (RNN) algorithm for an IDS on IoT data. The XBoost model was used for a min–max scaling to pre-process the datasets before feature selection. The findings show that the suggested method is efficient in classifying the KDD and CICIDS-2017 datasets and achieves an accuracy of 99% on DDoS attacks.

Similarly, Balega et al. [16] utilized ML models, such as XGBoost, SVM, and CNN, for anomaly detection in IoT. The results indicate that XGBoost demonstrated superior performance compared to both SVM and DCNN by attaining an accuracy of 99.98% and 99.84% on the IoT-23 and TON_IoT datasets, respectively, and with a training speed that was 717.75 times quicker than other implemented models. In the same way, Javed et al. [17] suggested implementing an innovative two-step IDS for IoT devices, utilizing ML techniques. This approach aims to improve both the precision and the computing efficiency of the system. The first layer of the suggested IDS is deployed on a microcontroller-based smart thermostat. The smart thermostat then sends the information it has gathered to a website that is stored on a cloud. The next tier is implemented to categorize and identify different types of attacks. The recommended technique is capable of identifying various kinds of attacks on a smart thermostat within a time frame of 3.51 milliseconds, with an accuracy rate of 97.59%.

Likewise, Yaras and Dener [18] intended to examine the acquired network traffic information within a big data context and identify network threats utilizing a deep learning technique. The correlation approach diminishes the dataset’s features, guaranteeing the retention of pertinent information in the analysis. A hybrid deep learning technique was developed utilizing a 1D CNN and LSTM. The study’s findings show that the ‘CICIoT2023’ dataset achieves an accuracy rate of 99.96% for multiclassification and 99.995% for binary classification. The accuracy obtained on the TON_IoT dataset was 98.75%. Similarly, Li et al. [19] developed the ML-based IDS by feature selection and extraction method on the TON-IoT network dataset and achieved the highest accuracy of 77.42%.

Correspondingly, Krishnan and P. Shrinath [20] aimed to detect attacks by using the stacked multi-classifier trained with class-specific features for the identification of known assaults. To tackle the issue of significant class imbalance without employing resampling techniques, the Localized Generalized Matrix Learning Vector Quantization (LGMLVQ) method was implemented to identify the pertinent features for each class. The highest accuracy achieved was 93.715% on the NF-ToN-IoT dataset. Awotunde et al. [21] employed hybrid models integrated with a selection of features in IoT networks. The recommended approach utilizes Chi-Square to select the features. Simultaneously, multiple classifiers, such as XGB, Bagging, ET, RF, and AdaBoost, were employed for intrusion detection on the IoT_Telemetry datasets. The accuracy achieved on the network datasets was 99.09%, while the maximum accuracy on IoT datasets was 99.73%. Similarly, Alotaibi and Ilyas [22] developed an ensemble model with DT, RF, logistic regression, and KNN using stacking and voting techniques. The results indicate that the framework enhances the performance of the IDS with an accuracy of 98.63% on the network dataset.

Arreche et al. [23] suggested the use of XAI techniques with seven different traditional ML models and a DNN model with the use of three IDS datasets, i.e., RoEduNet-SIMARGL2021, CICIDS-2017, and KDD. The primary focus of the study is the use of XAI for features. In addition, feature extraction methods are used to identify important aspects that are specific to the model and the intrusion, which helps us understand the factors that discriminate and impact the detection results. The recommended technique also finds overlapping and relevant factors that affect many AI models, revealing shared patterns across detection methods. The findings suggest that in the majority of AI models, the processing overhead required to generate XAI explanations is negligible, guaranteeing usefulness in real-time situations. However, this study only focuses on ML techniques and DNNs, not highlighting the importance of using the XAI in DL.

The literature analysis suggests that most of the recent research focuses on using machine learning algorithms like DT, RF, XGBoost, SVM, KNN, etc. and DL algorithms like CNN (DenseNet, ResNet, VGG, Inception v3, etc.) Also, many researchers use ensemble learning, like stack classifiers, which combine two or more classifiers to enhance the performance of the model. Among all the approaches, in most of the studies, the highest performing model is the boosting algorithm, XGBoost, and the ensemble learning classifier, which is proven to give the best performance. Also, most of the studies focus on binary classification. Thus, there is a need to improve the research by working on such datasets where five–eight different types of attacks are present to classify. Also, the limitation of current studies is that no study has explained the interpretation of their results for DL-based IDSs. As ML is the black box, the interpretation of results and its explainability are essential to make the model trustworthy in real-time applications. Hence, this paper proposes a novel DL-based trustworthy IDS framework by developing customized CNN and DNN architectures and comparing their performance with the transfer learning model TabNet. Also, the XAI techniques are implemented to explain the predictions made by the models and to find the most influencing features responsible for predicting the class in multi-class classification.

## 3. Materials and Methods

This paper aims to identify different types of attacks from network and IoT datasets. For this purpose, the proposed method develops a novel DL algorithm like DNN and 1D-CNN. The proposed architecture is given in Figure 1.

### 3.1. Dataset Description

The seven different datasets are from TON_IoT, UNSW [24] research. The ToN_IoT datasets constitute a novel generation of Industry 4.0/Internet of Things (IoT) and Industrial IoT (IIoT) data collections intended for assessing the efficacy and precision of diverse cybersecurity solutions, especially those leveraging AI, including ML and DL algorithms. The datasets are referred to as ToN_IoT due to their integration of data from several sources, encompassing telemetry data from IoT and IIoT sensors, operating system logs from Windows 7, Windows 10, Ubuntu 14, and Ubuntu 18 TLS, in addition to network traffic data. The data are obtained from a private network created at the Cyber Range and IoT Labs at UNSW Canberra, Australia.

A testbed network was established to emulate Industry 4.0, encompassing IoT and IIoT networks. This network was created with many virtual machines and hosts executing on Windows, Linux, and Kali Linux, enabling the combination of IoT, Cloud, and Edge/Fog layers. The dataset includes various assaults, such as DoS, DDos, and ransomware, targeting web applications, IoT gateways, and computer systems within the IoT/IIoT network. The data-collecting operation was conducted concurrently, acquiring both benign and malicious activities from network traffic, Windows and Linux audit logs, and IoT telemetry data.

The various versions of the datasets are available: processed, raw, train-test, and many more. For this research, the train-test version of the datasets was used. The seven datasets used are one network and six datasets of IoT/IIoT, i.e., IoT_Fridge, IoT_Garage_Door, IoT_GPS_Traker, IoT_Modbus, IoT_Motion_Light, and IoT_Therostat.

### 3.2. Data Pre-Processing and Encoding

All the datasets are pre-processed to find missing values and duplicates. As the network dataset is quite large and has different variations, scaling is performed to normalize the values in a similar range for better optimization [25]. Especially in the case of deep learning algorithms that use the vanishing gradient approach, it allows for a faster convergence by removing the impact of significant disparities in feature scales on the optimization process. Normalization improves model consistency and accuracy by ensuring that all features contribute equally in terms of training the model and propagating errors. Moreover, it enhances numerical stability in algorithms that involve matrix operations and guarantees the consistent application of regularization techniques across features, thereby mitigating the risk of overfitting [26].

Label encoding is used to encode the features and labels of the datasets. As the network dataset consists of 43 features, one-hot encoding would increase the features, which results in a longer time for training the models [27]. Hence, due to the resource constraints, this paper used label encoding to keep the same number of features even after encoding.

### 3.3. Deep Learning Models

Two customized deep learning models and one transfer learning model, i.e., TabNet [28], were developed and used to train on the pre-processed, encoded, and normalized datasets individually. For the IoT_Modbus dataset, the architecture of the proposed CNN is given in Figure 2, and for the DNN, it is given in Figure 3. The model architectures for all the datasets are the same; only the number of classes in the last dense layer changes as per the number of categories in the label for each dataset. For the IoT_Modbus dataset, the number of neurons is 6, as the number of categories in classes is 6.

As shown in Figure 2, the CNN architecture consists of various layers. It consists of four layers: a convolution layer, a pooling layer, a Dropout layer, and a fully connected layer, i.e., a dense layer. These layers are stacked to form the CNN architecture [29]. Equation (1) determines the equation for calculating the output of the convolution layer with size *N*.(1)$$N=\frac{W-F+2P}{S}+1,$$
where *N* = the size of the output feature map, *W* = the length of the input sequence, *F* = the size of the kernel/filter, *P* = the amount of padding, and *S* = stride

Pooling layers diminish the dimensions of the feature map, facilitating the down-sampling of the input representation, easing computational demands, and mitigating overfitting. The most prevalent method of pooling is Max Pooling, which retrieves the maximum value from an area of the input feature map [30]. The formula for determining the output of the max pooling layer is given in Equation (2).(2)$$N=\frac{W-F}{S}+1,$$
where *N* = the output size after pooling, *W* = the input size before pooling, *F* = the pool size like a filter in convolutional operation, and *S* = the stride used in the pooling layer.

Dropout does not alter the dimensions of the input or output. To avoid overfitting, it randomly eliminates (zeroes out) a percentage of activations during training. The output dimension is identical to the input dimension [31]. If the input size is $N_{input}$, then the formula for the output of the dropout layer is given in Equation (3).(3)$$N_{dropout}=N_{input},$$

For the dense layer, multidimensional inputs need to be flattened. The output of the dense layer is determined by the number of neurons in the dense layer [32]. The last dense layer, which is responsible for the classification and prediction of the class, consists of a number of neurons similar to the number of classes in the target variable. If the flattened input size is $N_{input}$ and if the number of neurons in the dense layer is U, then the formula for the dense layer’s output is given in Equation (4).(4)$$N_{Dense}=U,$$

As shown in Figure 3, the DNN architecture consists of a dense layer and a dropout layer like the CNN architecture; only the added layer is batch normalization. Batch normalization is a deep learning method that enhances the training of deep neural networks by normalizing inputs to layers during mini-batch training. It decreases problems associated with internal covariate shifts, thereby enhancing the speed and stability of training [33]. The equation for batch normalization is given in Equation (5).(5)$${\hat{o}}_{b,n}=\gamma_{n}\cdot (\frac{I_{b,n}-\mu_{n}}{\sqrt{\sigma_{n}^{2}+\epsilon }})+\beta_{n}$$
where ${\hat{o}}_{b,n}$ = the normalized output of the bth sample in the batch at nth neuron; $I_{b,n}$ = the input to the batch normalization layer for bth sample in the batch at nth neuron; $\mu_{n}$ = the mean of the nth neuron across the mini-batch; $\sigma_{n}^{2}$ = the variance of the nth neuron across the mini-batch; $\gamma_{n}$ = the scaling parameter for the nth neuron, which is learned during training; ϵ = the constant for numerical stability; and $\beta_{n}$ = the shifting parameter for the nth neuron.

The selection of customized DL models is based on multiple experiments performed on different ML and DL models with different variations in model architecture. However, the most promising results were for the 1D-CNN, the DNN, and TabNet with the optimized architecture defined in the paper in Figure 2 and Figure 3, respectively. Hyperparameter tuning was performed to find the best hyperparameters for both customized models.

### 3.4. Transfer Learning Model: TabNet

The pre-trained model TabNet [28] is used in this research to identify the different types of intrusions in network and IoT datasets. It is a deep learning architecture specially used for tabular data. Google Cloud AI researchers introduced it and integrated deep learning methodologies with decision tree-like frameworks to manage the distinctive attributes of tabular data adeptly. The most crucial feature of TabNet is the sparse attention mechanism, which enables the model to concentrate on a specific subset of features during each decision step. This method improves interpretability relative to conventional neural networks by emphasizing the features most pertinent to each prediction. The architecture separately identifies relevant features while disregarding extraneous ones, minimizing noise from irrelevant data and enhancing generalization and performance. TabNet also uses multi-head attention to develop various pathways for data processing, allowing the model to learn from diverse facets of the feature set concurrently. This capability guarantees the effective capture of multiple interactions within the data. Moreover, the architecture is constructed to yield interpretable predictions, facilitating practitioners’ comprehension of the impact of individual features on the output. TabNet’s architecture facilitates sequential decision-making, allowing the model to discern intricate interactions within the data and enhancing its overall efficacy in managing tabular data.

TabNet was selected to be compared with the customized CNN and DNN models as the TabNet transfer learning model is the deep learning model designed for training on structured datasets. The TabNet is not customized, as the objective of this paper is to test the pre-trained model directly on the new dataset, i.e., the datasets used for the experiments to verify its performance by just fine-tuning the hyperparameters.

### 3.5. XAI

All the models utilized in this study are black-box models; hence, the reason why a particular prediction is made is unknown. Thus, to build the trust of the network administrators in the proposed solution, it is necessary to explain the predictions and which features are responsible for predicting each particular class. Thus, this research utilizes the two XAI techniques, i.e., LIME and SHAP, to explain the predictions of the models.

The LIME technique [34] approximates the model locally around the instance that needs to be explained in order to explain individual predictions. To determine which features are most important for a given prediction, perturbed samples around the instance are created, and an interpretable model (like linear regression) is fitted to these samples. However, SHAP [35] uses cooperative game theory as its foundation and calculates SHAPley values for every feature in a model. This method ensures consistent and theoretically supported explanations by considering all possible feature combinations to distribute each feature’s contribution to a prediction equitably.

LIME and SHAP are the most commonly used techniques for explaining the predictions in structured datasets, as suggested by Gaspar et al. [36] and Sharma et al. [37]. They are especially favorable methods for cybersecurity applications [38]. As all the datasets used for this research are structured, LIME and SHAP were selected and used. The LIME results were calculated by perturbing the input instance, predicting outputs utilizing the model, fitting a surrogate linear model to predictions, and analyzing feature weights to determine importance as given in Algorithm 1. However, SHAP results were computed first to generate feature subsets, predict outputs for each subset, compute SHAP values representing each feature’s contribution, and identify feature importance based on their SHAP magnitudes as given in Algorithm 2.


**Algorithm 1: LIME**

1.Let $x\in R^{d}$ be the instance from the dataset that needs to be explained.2.Generate permutations {$x_{i}$} such that $x_{i}=x+\epsilon_{i}$, where $\epsilon_{i}$ is delta changes to *x* for variations, for *i* = 1, 2, …, m.3.Predict the output of the perturbed instances using the model *f* as ${\hat{y}}_{i}=f(x_{i})$, for *i* = 1, 2, …, m.4.Fit the surrogate model g to the predictions as $g\left(x\right)=\sum_{j=1}^{d}\omega_{j}x_{j}+b$5.Minimize the difference between $g\left(x_{i}\right)$ and ${\hat{y}}_{i}$ by $\underset{g}{min}\sum_{i=1}^{m}{\left(g\left(x_{i}\right)-{\hat{y}}_{i}\right)}^{2}$6.Generate the explanations by analyzing the weights $\omega_{j}$ to determine the importance of the features as features with higher $\left|\omega_{j}\right|$ are considered more important.



**Algorithm 2: SHAP**

1.Let $x\in R^{d}$ be the instance from the dataset that needs to be explained.2.Let $f\left(x\right)$ be the trained DL model. It gives the prediction, for instance *x*.3.Create the subset *S* of the features, where each subset *S* consists of some features of *x* and the remaining features are assigned to the baseline value.4.For each of the subset *S*, predict the classes for the model $f\left(x_{s}\right)$, where $x_{s}$ is the instance for the features in the subset *S*.5.Calculate SHAPley value ($\emptyset_{j}$) for the feature *j* as $\emptyset_{j}=Average\;contribution\;of\;j\;to\;the\;model\;prediction.$6.The final prediction $f\left(x\right)$ is evaluated as $f\left(x\right)=Baseline\;prediction+\sum_{j=1}^{d}\emptyset_{j}$.7.Calculate feature importance: features with a high value of $\left|\emptyset_{j}\right|$ are more important to the prediction of the model.


### 3.6. Performance Measures

The standard performance metrics used for evaluating the classification models are given below. The formulas for accuracy, precision, recall, F1 score, and cross-entropy loss are shown in Equations (6)–(10).

Accuracy is the proportion of cases that were accurately predicted in all instances. If the dataset is unbalanced, it could be misleading [39].(6)$$Accuracy=\frac{True\;Positives+True\;Negatives}{Total\;instances}$$

Precision calculates the percentage of positively predicted cases that were accurately predicted out of all positively predicted instances. When false positives are essential, it is helpful [39].(7)$$Precision=\frac{True\;Positives}{True\;Postives+False\;Positives}$$

Recall shows the percentage of real positive cases that the model accurately predicted. It is helpful when there is a possibility of false negatives [39].(8)$$Recall=\frac{True\;Positives}{True\;Postives+False\;Negatives}$$

The F1 score is the harmonic mean of the Precision and Recall. It is helpful in situations where there is an unequal distribution of classes because it offers a single metric that addresses both issues [39].(9)$$F1\;score=2\times \frac{Precision\times Recall}{Precision+Recall}$$

Cross Entropy loss/Log loss calculates the degree of prediction uncertainty. It is an effective metric for probabilistic classification because it penalizes incorrect predictions with probabilities that are far from the actual values [40].(10)$$Log\;loss=-\frac{1}{N}\;\sum_{i=1}^{N}\sum_{c=1}^{C}y_{i,c}\;\log(p_{i,c})$$

Receiver operating characteristic—area under the curve (ROC-AUC) represents the model’s capacity to discriminate between classes at all threshold values. A higher AUC indicates more excellent model performance [39].

## 4. Results and Discussion

The experiments were performed using Python on the Kaggle platform using Titan X P100 Graphical processing units. The pre-processed and normalized data were trained on the three different DL models. The transfer learning model, TabNet, is the most famous model for training on structured data. The two customized CNN and DNN architectures were developed and trained on the pre-processed encoded data. The comparison of the three proposed algorithms on all the datasets is given in Table 1. The findings suggest that the network dataset achieves the highest accuracy of 99.24% for the CNN model. In comparison, the average accuracy of the test data for the IoT datasets is 100%, as all the datasets achieved more than 99.91% accuracy. In most cases, the CNN outperforms the other two models, while with the IoT-Fridge, the DNN model outperforms by achieving 0.07% more accuracy than the CNN.

### 4.1. Network Dataset

The training and validation accuracy and loss graphs for all the datasets are given below. For the network dataset, the graphs are given in Figure 4 and Figure 5 for the CNN and the DNN, respectively. The ROC curves for the CNN, the DNN, and TabNet are depicted in Figure 6, where 0 represents a backdoor, 1 corresponds to a DDoS, 2 to a dos, 3 to an injection, 4 to a mitm, 5 to a typical, 6 to a password, 7 to ransomware, 8 to scanning, and 9 to xss.

Figure 4 shows that the training and validation accuracy increases drastically by the 10th epoch while increasing very slowly up to the 50th epoch. The opposite trend is observed for the losses. Finally, early stopping results in the highest accuracy at the 50th epoch. Alternatively, the DNN model’s accuracy increases drastically until the 40th epoch while becoming steady and finally achieving the best performance at the 56th epoch. The ROC curves for the network dataset show that all the classes are classified with 1 area under the curve (AUC), while only two classes, 3 and 6, have 0.99 AUC, as shown in Figure 6b for TabNet. The injection and password classes have less accuracy than the other classes classified by the TabNet model.

Figure 7 shows the eXplainable AI results for the network dataset generated by the DNN model. In this situation, the model predicts that the input instance belongs to the NOT 1 class, i.e., DDos with a 1.00 probability. The left side of the figure depicts the prediction probabilities for various courses, while the right side lists the features and their corresponding values for the input instance. As the features are normalized, the values of all the features are 0. The colorful bars on the left side indicate how important each feature is in contributing to the prediction, with the longer bars having more importance, like dst_bytes, having an importance value of 0.75.

The performance evaluation on test data of all three models on the network dataset is given in Table 2.

### 4.2. IoT-Fridge Dataset

Figure 8 shows the training and validation graphs for the CNN, and Figure 9 shows the training and validation graphs for the DNN for the IoT-Fridge dataset. The training and validation graphs show that the accuracy dramatically increases in the first 5 epochs and then increases very slowly for both the CNN and the DNN. The CNN achieves the highest accuracy at the 40th epoch, and the DNN achieves the highest performance at the 35th epoch. The ROC curves are the same for the CNN, the DNN, and TabNet, as shown in Figure 10, where 0 represents the class backdoor, 1 is DDos, 2 is injection, 3 is normal, 4 is password, 5 is ransomware, and 6 is xss. The ROC curve for the classes is at point (0,1) and with AUC 1; hence, only the single graph is visible as all coincide with each other.

The performance of all three models is compared with different performance metrics, as given in Table 3. The DNN achieves the highest performance among the three models with 99.99% accuracy.

The LIME results depicted in Figure 11 elucidate the forecast rendered by a DNN for categorizing a data point as an injection. The model forecasted injection with absolute certainty, as seen by the probability bar, while alternative classifications, e.g., DDos and normal, exhibited a 0% probability. Important features comprise date, time, temp_condition, and fridge_temperature, with their respective values influencing the prediction’s inclination towards DDos or NOT DDos. The orange bars emphasize traits that contribute to the positive class (DDos), whereas the blue bars denote those that promote the negative class. The feature importance bar on the right displays the values and effects of each feature on the prediction, with fridge_temperature and date exerting the most substantial influence on the model’s decision. The LIME results employ comparative labels to elucidate the impact of feature values on the predictions of the DNN model. The characteristic data less than equal to −0.85 significantly aids in predicting DDos, as values of the data that are less than or equal to −0.85 strongly indicate this class. The feature importance score is 0.41. Likewise, temp_condition less than equal to −0.98 signifies that diminished values of temp_condition further incline the model to forecast DDos. Conversely, the attribute time has a more intricate function, with values ranging from −0.87 to −0.02, reinforcing the NOT DDos classification. These comparisons indicate the feature value ranges that exert the most significant influence on the model’s decision-making, with the positive class denoted in orange and the negative class in blue.

The SHAP results, shown in Figure 12, illustrate the contribution of individual features to the DNN model’s prediction of the injection class. Both the actual and anticipated labels are injections, with the predicted probability indicating high confidence in this classification (approaching 1, as demonstrated in scientific notation). The SHAP value plot below demonstrates the influence of each attribute on the model’s decision-making process. The base value, denoted as 0.00, signifies a neutral starting point. Features on the left side (in red) increase the likelihood of predicting injection, but those on the right side (in blue) diminish this likelihood. The label attribute, valued at 0.4468, significantly influences the prediction towards injection. The time variable, with a coefficient of −0.1071, exerts a moderate influence in the same direction. The date attribute, exhibiting a SHAP value of −0.8501, detracts from the prediction of injection, hence diminishing the probability of this classification. The model integrates the cumulative impact of these feature contributions to reach its definitive conclusion, resulting in a reliable prediction of injection.

### 4.3. IoT-Thermostat Dataset

Figure 13 and Figure 14 show the training and validation graphs for the CNN and DNN, respectively, for the IoT-Thermostat dataset. The ROC curve for all three models is the same as all the classes that have achieved an AUC of 1, which is similar to the ROC curve of the IoT_fridge dataset, as given in Figure 10.

The accuracy of the CNN and DNN models drastically increased from 0 to the 6th epoch, steadily increasing until they achieved the highest performance at the 42nd for the CNN and the 30th for the DNN. The performance evaluation of all three models is given in Table 4. Among all of them, the CNN and DNN outperformed by achieving 100% accuracy.

The LIME results of the CNN, as shown in Figure 15, also show that time is the critical feature in detecting whether the instance belongs to the class ‘injection.’ For this dataset, as there is less data, the XAI results for the two different methods are generated. The SHAP plot, as depicted in Figure 16, shows how a CNN model arrived at its prediction for a given instance. In this scenario, the model was assigned to determine whether a condition labeled injection existed. The base value, which represents the model’s expected output if none of the input features were included, is depicted in the center of the plot. This base value is set to zero, indicating a neutral start point. The distinct bars that extend to the left and right sides of this base value show the tendency of the feature toward or away from the injection label. Red bars indicate negative contributions, which reduce the prediction probability, whereas blue bars indicate positive contributions, which improve the probability.

The most relevant variable in this prediction was time, which had a strong positive impact. This shows that the model largely depended on the time characteristic to make the forecast, closely linking it with the occurrence of an injection. In contrast, the current_temperature feature had a negative SHAP value of −0.4498, indicating that it somewhat shifted the prediction away from injection. Similarly, the date feature had a negative impact, with a SHAP value of −0.4762, shifting the prediction away from the target label. However, like current_temperature, its effect was minor. Finally, the thermostat_status feature had a lesser positive contribution, with a SHAP value of 0.3513.

### 4.4. IoT-MotionLight Dataset

The CNN training graph is in Figure 17, whereas the DNN validation graph is in Figure 18. The models’ highest performances were achieved at the 54th and 67th epochs, respectively. The ROC curves for the three models are straight lines pointing to the point (0,1), as all the models achieved AUC 1 for all the classes. Table 5 compares all the models using different performance metrics. Among the three models, the CNN outperforms the DNN and TabNet by achieving an accuracy of 99.94%. The XAI results of the highest-performing CNN model are given in Figure 19. The SHAP plot indicates that the parameter time significantly influences the prediction towards injection with a value of 1.59, but characteristics such as date, light_status, and motion_status diminish the probability of predicting injection. The model accurately categorizes the data as injection with complete certainty, as seen in the prediction probabilities. The LIME section elaborates on the contribution of features in differentiating between DDos and NOT DDos scenarios. All the features have a feature importance of 0 for the given example.

### 4.5. IoT-Garage Dataset

The graphs for the CNN are given in Figure 20, whereas those for the DNN are shown in Figure 21. The models’ highest performances were achieved at the 50th and 37th epochs, respectively. The ROC curves for the three models are straight lines pointing to the point (0,1) as all the models achieved AUC 1 for all the classes, as given in Figure 10. Table 6 compares all the models using different performance metrics. Among the three models, the CNN achieves 99.92% accuracy, which outperforms the DNN and TabNet.

The XAI results of the CNN are given in Figure 22. The SHAP and LIME analyses reveal that date having a value of 1.07 favorably impacts ransomware prediction but has no impact on predicting the NoT DDos class, and the same is true for time having a value of −1.16. Attributes such as door_state and sphone_signal do not influence the classification of the class DDos, with their values influencing the outcome.

### 4.6. IoT-Modbus Dataset

The graphs for the CNN are shown in Figure 23, and those for the DNN are given in Figure 24. The models’ highest performances were achieved at the 32nd and 76th epochs, respectively. The ROC curves for the three models are straight lines pointing to the point (0,1) as all the models achieved AUC 1 for all the classes, as given in Figure 10. Table 7 compares all the models using different performance metrics. All three models achieve an accuracy of 100%. The XAI results of the DNN are given in Figure 25. Time and date are the most influential features with a feature importance of 0.29 and 0.28, respectively, for the prediction of the normal class in a given example.

### 4.7. IoT-GPS Dataset

The graphs for the CNN are given in Figure 26, while those for the DNN are shown in Figure 27. The models’ highest performances were achieved at the 32nd and 76th epochs, respectively. The ROC curves for the three models are the same, as all the models achieved 1 AUC for all the classes, as shown in Figure 10 for the IoT_Fridge dataset. Table 8 compares all the models using different performance metrics. The CNN and DNN models achieve an accuracy of 100%. The XAI results of the CNN are given in Figure 28, which indicates that date, time, and longitude are the most influential features for predicting the given instance.

By exceeding previous methods compared to the same dataset, the suggested approach achieves an average accuracy of 99.96% on test data on the IoT datasets and a 99.24% accuracy on the network dataset, as shown in Table 9.

The proposed solution tried to resolve the overfitting issues in the proposed models by utilizing batch normalization and dropout layers. Also, the dataset is pre-processed to achieve the highest performance. Still, the difference between the test accuracy and the training accuracy for the network dataset is just 0.74% for the CNN, and on the IoT dataset, it is 0.04% for the CNN. So, the problem is not with the model’s architecture but with the dataset. As in the case of the IoT datasets, 100% accuracy is achieved for four datasets, while, the accuracy achieved by the proposed CNN model for the IoT_motionlight dataset is 99.5 and the accuracy for the IoT_garage_door dataset is 99.1.

## 5. Conclusions

A novel intrusion detection system architecture is presented in this paper. This architecture makes use of DL and XAI techniques to generate explainable models for network intrusion detection systems. This architecture gives security analysts the ability to use trustworthy IDSs. In order to categorize the intrusions, three distinct DL models were created, i.e., the CNN, the DNN, and the transfer learning model TabNet. Seven different open-source datasets from TON-IoT were used to evaluate the models’ efficiency. One dataset is a network dataset, while the other six are IoT datasets. An accuracy of 99.24% on the test dataset was attained by the CNN model when applied to the network dataset. Concurrently, the CNN and the DNN achieved 100% accuracy in the majority of the six IoT datasets. When compared across all datasets, TabNet performed the worst. An explanation of the predicted outcomes is necessary for putting the suggested method into action in real time. Consequently, XAI methods, i.e., LIME and SHAP, are used to comprehend the key attributes that are accountable for class prediction. This proposed IDS is a trustworthy system to be implemented by the network security analyst in real-time, which can also define the essential attributes that are responsible for predicting the particular instance of the data as malicious.

## Figures and Tables

**Figure 1 sensors-25-00847-f001:**
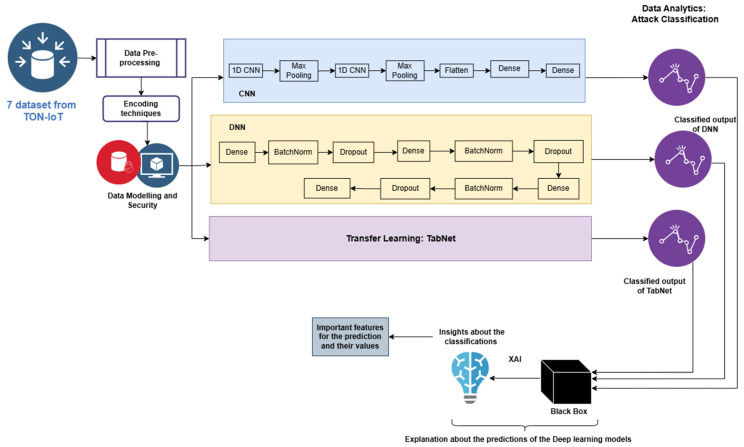
XAI and DL-based architecture for attack classification.

**Figure 2 sensors-25-00847-f002:**
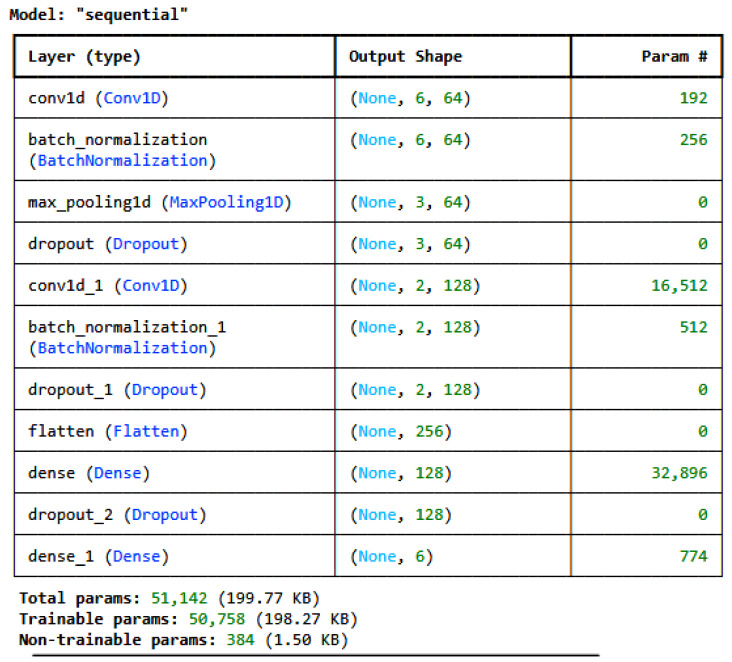
CNN architecture for the IoT_Modbus dataset.

**Figure 3 sensors-25-00847-f003:**
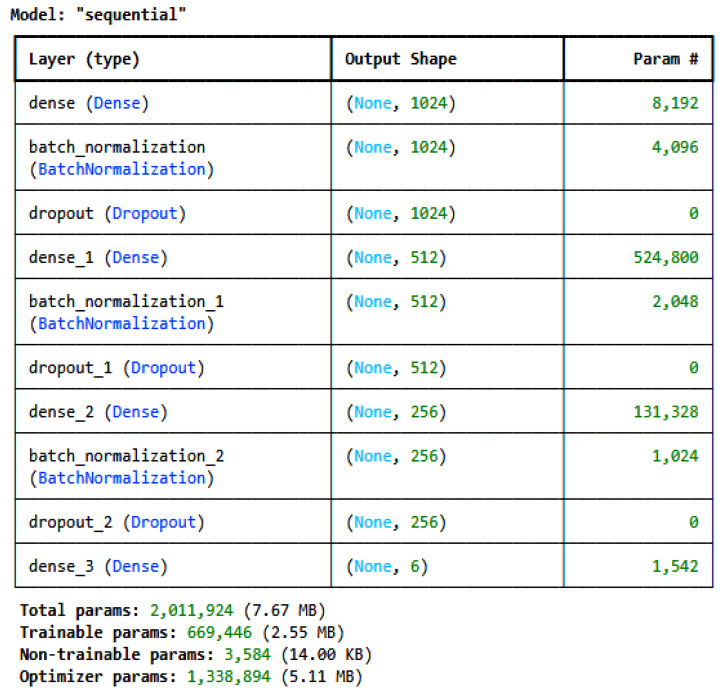
DNN architecture for IoT_Modbus dataset.

**Figure 4 sensors-25-00847-f004:**
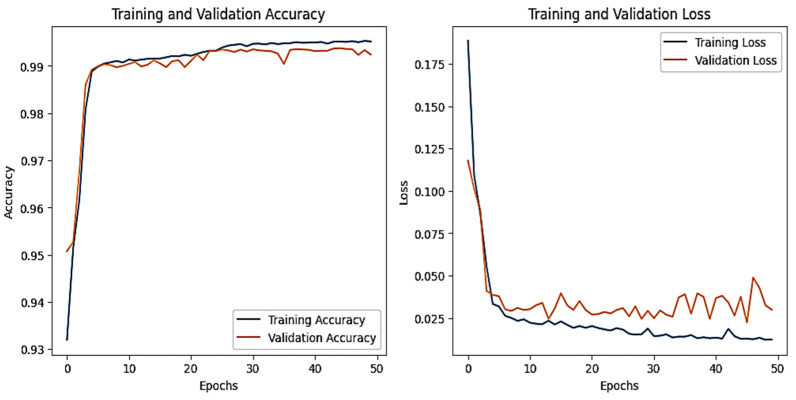
CNN–Network dataset’s accuracy and loss graphs.

**Figure 5 sensors-25-00847-f005:**
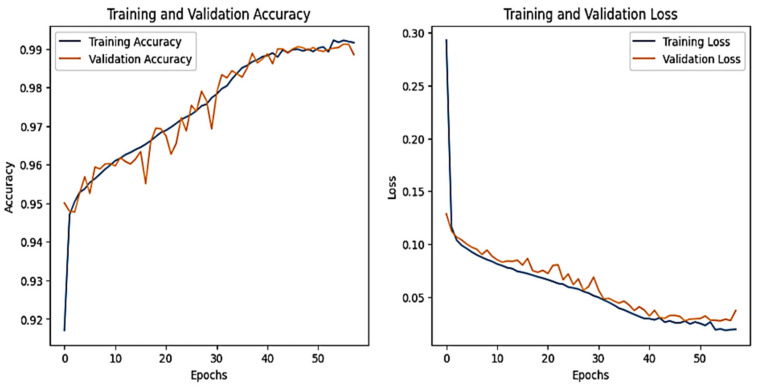
DNN–Network dataset’s accuracy and loss graphs.

**Figure 6 sensors-25-00847-f006:**
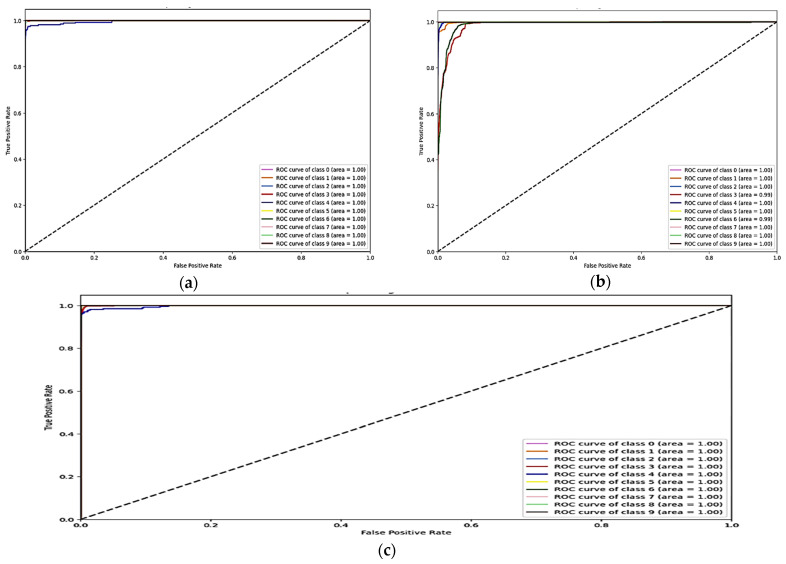
ROC curve for the network dataset (**a**) CNN, (**b**) TabNet, and (**c**) DNN.

**Figure 7 sensors-25-00847-f007:**
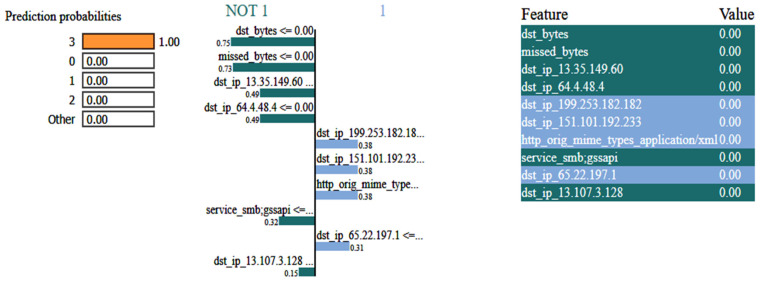
LIME results for the network dataset.

**Figure 8 sensors-25-00847-f008:**
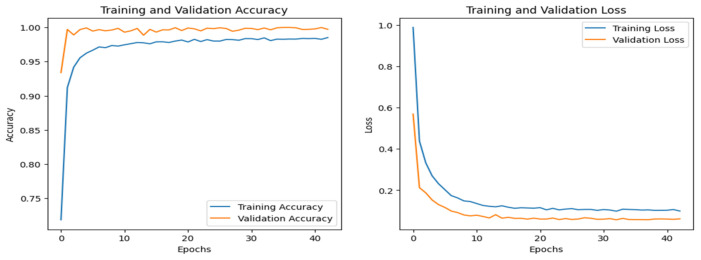
CNN–IoT-Fridge dataset’s accuracy and loss graphs.

**Figure 9 sensors-25-00847-f009:**
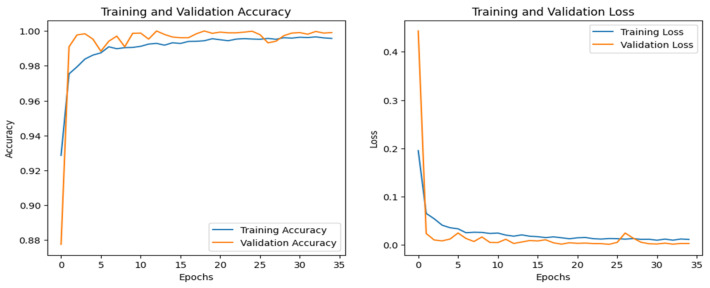
DNN–IoT-Fridge dataset’s accuracy and loss graphs.

**Figure 10 sensors-25-00847-f010:**
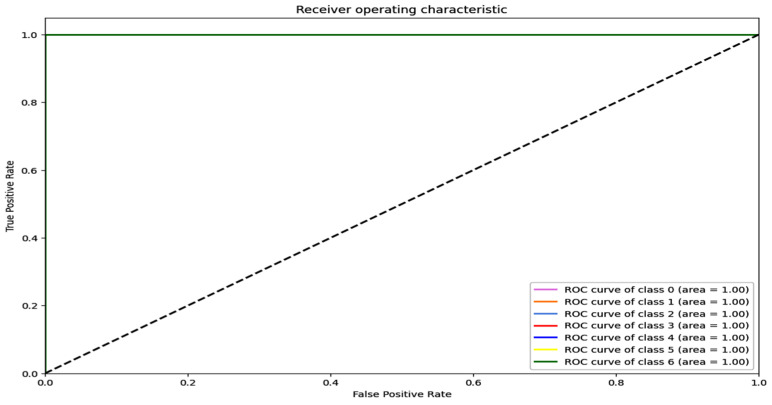
ROC curve for CNN, DNN, and TabNet for IoT-Fridge dataset.

**Figure 11 sensors-25-00847-f011:**
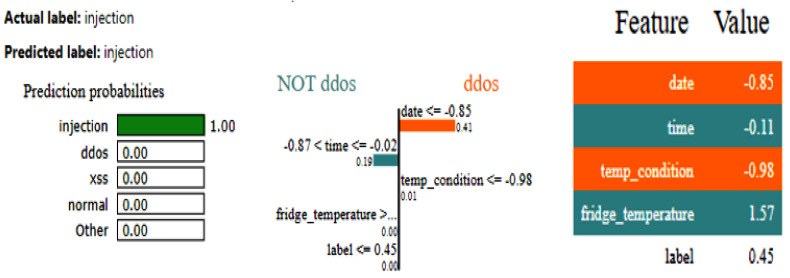
LIME results for the DNN model for the IoT_Fridge dataset.

**Figure 12 sensors-25-00847-f012:**
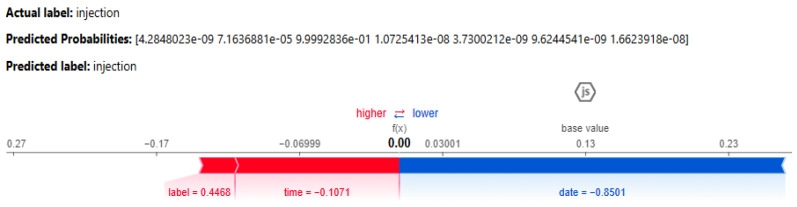
SHAP results for the DNN model for the IoT_Fridge dataset.

**Figure 13 sensors-25-00847-f013:**
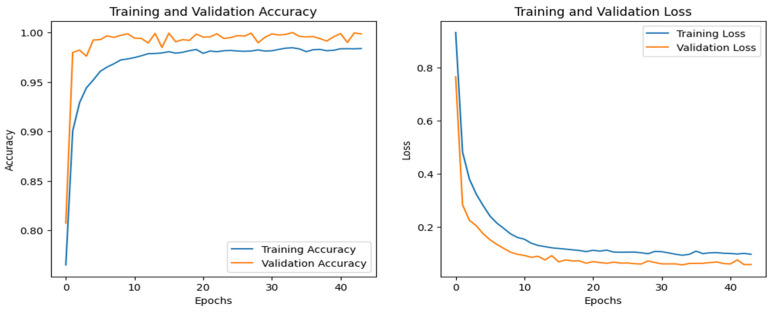
CNN–IoT-thermostat dataset’s accuracy and loss graphs.

**Figure 14 sensors-25-00847-f014:**
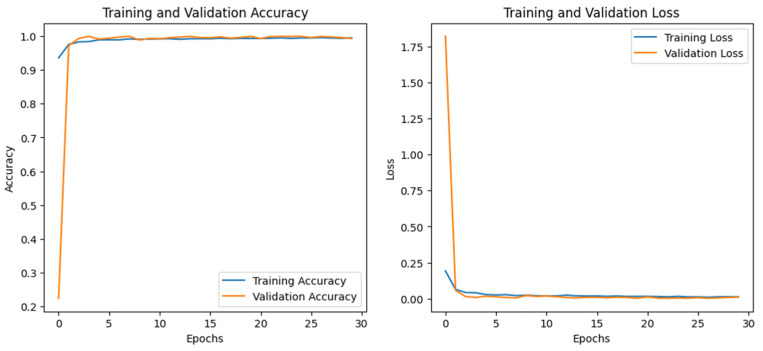
DNN–IoT-Thermostat dataset’s accuracy and loss graphs.

**Figure 15 sensors-25-00847-f015:**
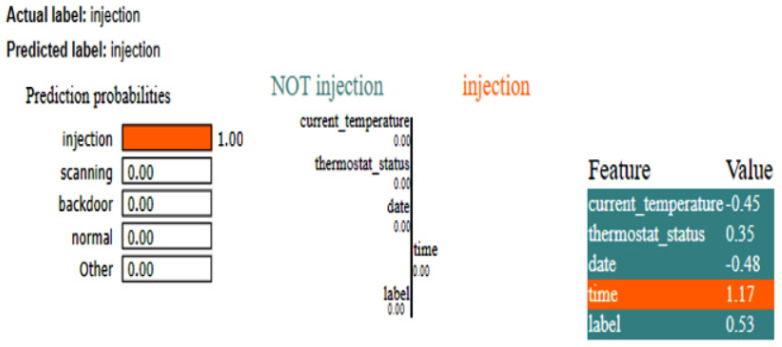
LIME results of CNN for IoT-Thermostat dataset.

**Figure 16 sensors-25-00847-f016:**
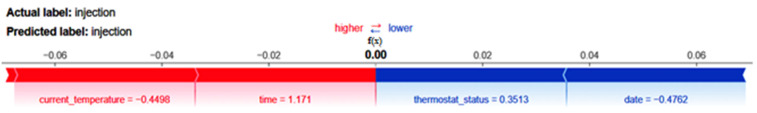
SHAP results of CNN for IoT-Thermostat dataset.

**Figure 17 sensors-25-00847-f017:**
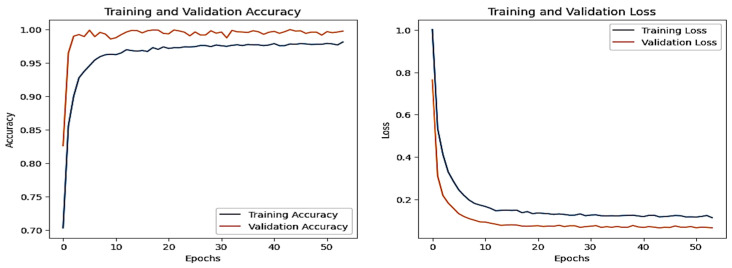
CNN–IoT-MotionLight dataset’s accuracy and loss graphs.

**Figure 18 sensors-25-00847-f018:**
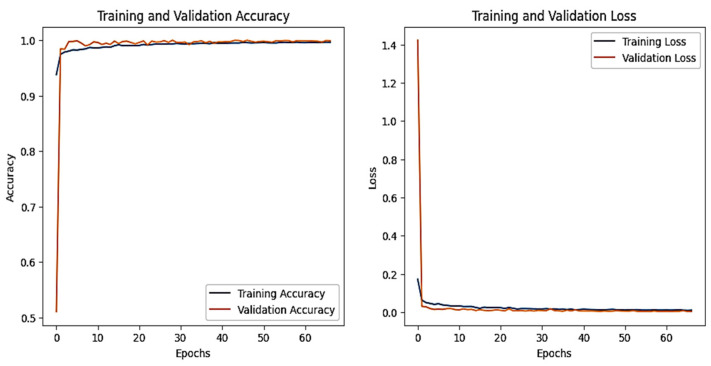
DNN–IoT-MotionLight dataset’s accuracy and loss graphs.

**Figure 19 sensors-25-00847-f019:**
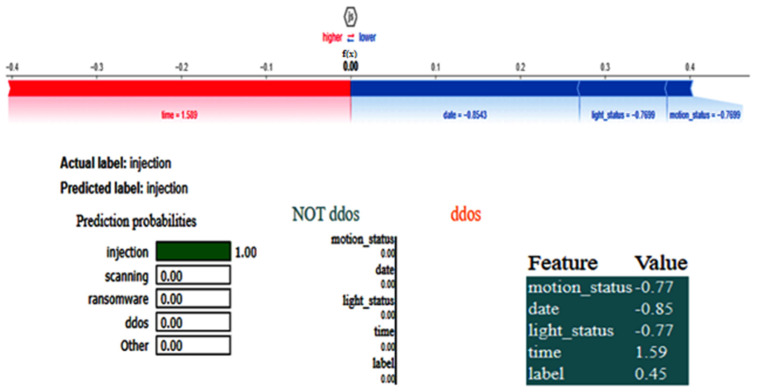
SHAP and LIME results for the CNN model for the IoT-MotionLight dataset.

**Figure 20 sensors-25-00847-f020:**
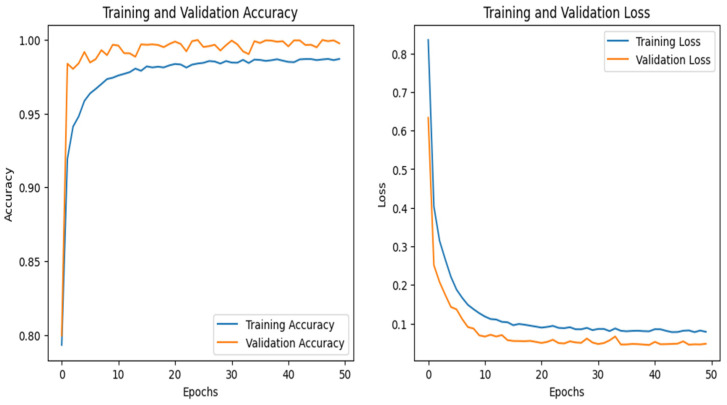
CNN–IoT-garage dataset’s accuracy and loss graphs.

**Figure 21 sensors-25-00847-f021:**
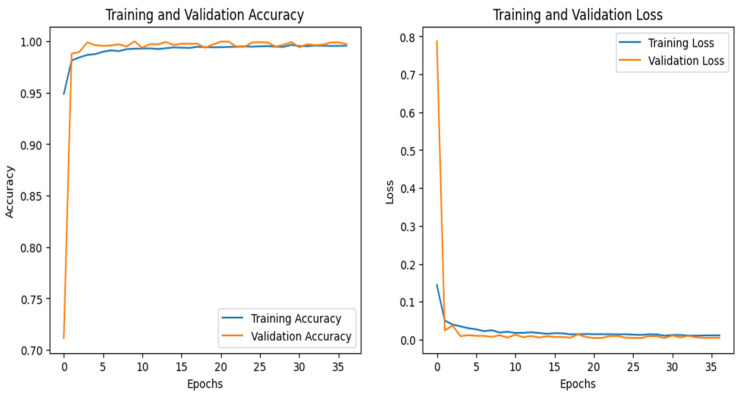
DNN–IoT-garage dataset’s accuracy and loss graphs.

**Figure 22 sensors-25-00847-f022:**
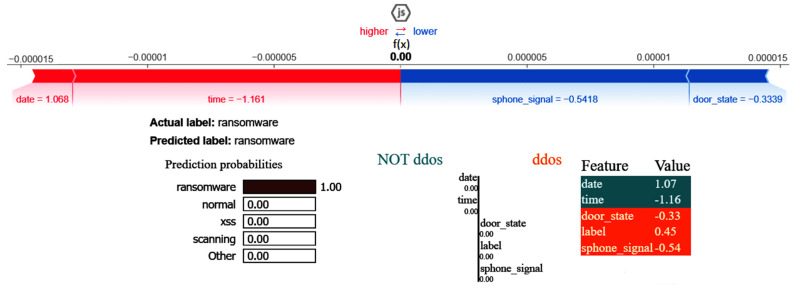
SHAP and LIME results of CNN.

**Figure 23 sensors-25-00847-f023:**
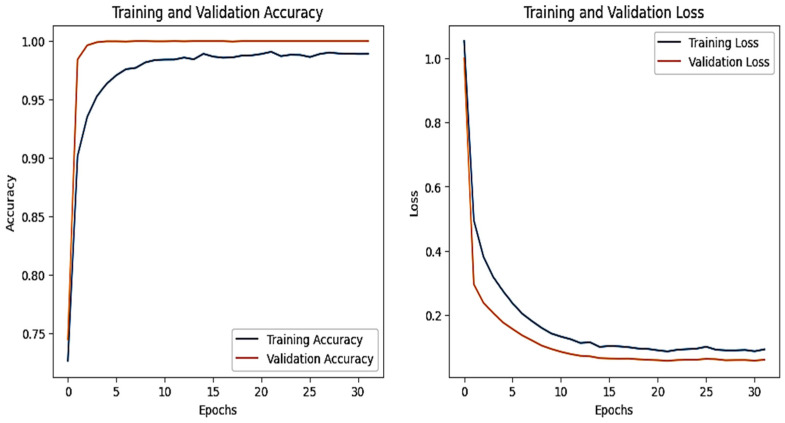
CNN–Iot-ModBus dataset’s accuracy and loss graphs.

**Figure 24 sensors-25-00847-f024:**
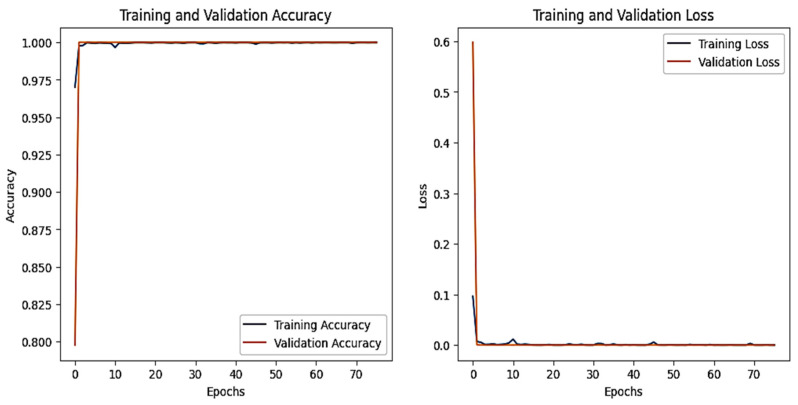
DNN–IoT-ModBus dataset’s accuracy and loss graphs.

**Figure 25 sensors-25-00847-f025:**
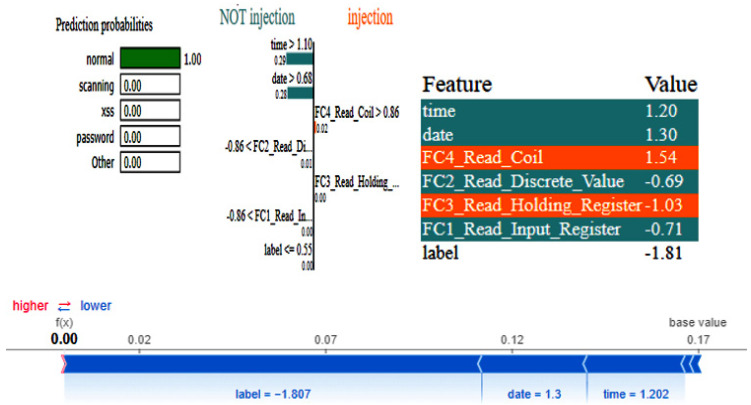
LIME and SHAP results of the DNN model.

**Figure 26 sensors-25-00847-f026:**
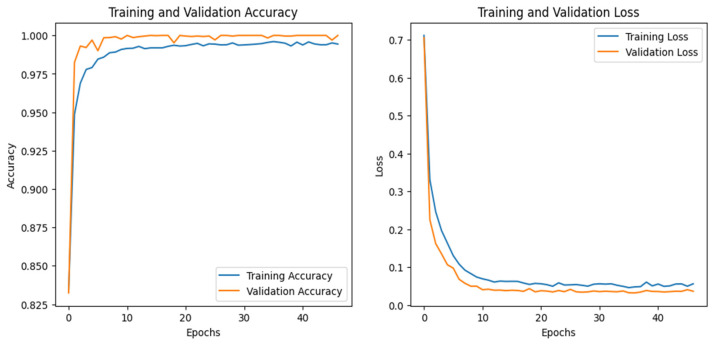
CNN–IoT-GPS dataset’s accuracy and loss graphs.

**Figure 27 sensors-25-00847-f027:**
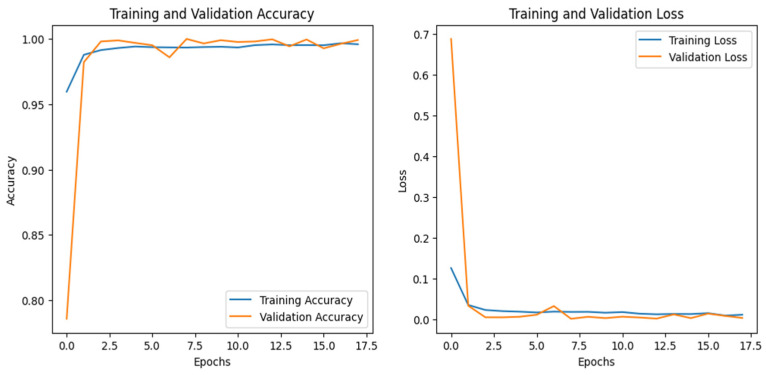
DNN–IoT-GPS dataset’s accuracy and loss graphs.

**Figure 28 sensors-25-00847-f028:**
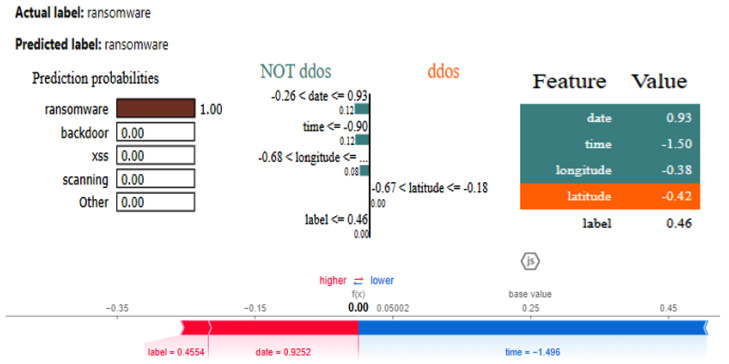
LIME and SHAP results of the DNN model.

**Table 1 sensors-25-00847-t001:** Comparison of all the algorithms with different datasets.

Dataset	Model	Accuracy			Loss			Epochs	Time
		Training	Validation	Test	Training	Validation	Test		(Seconds)
Network	CNN	99.53	99.24	99.24	0.0125	0.0299	0.0232	50 (50th)	1980
	DNN	99.02	98.86	99.03	0.0185	0.0371	0.0285	100 (56th)	276
	Tabnet	95.316	94.817	94.817	0.1279	0.1353	0.1353	2	72
IoT-Fridge	CNN	98.52	99.72	99.91	0.0985	0.0614	0.057	50 (38th)	43
	DNN	99.58	99.91	100	0.0107	0.003	0.0016	100 (34th)	35
	Tabnet	99.38	99.34	99.23	0.08907	0.08989	0.0899	5	82
IoT-Thermostat	CNN	98.37	99.86	100	0.0979	0.0576	0.0553	100 (44)	42
	DNN	99.59	99.33	100	0.0111	0.0116	0.0023	100 (30)	30
	Tabnet	93.13	91.24	98.21	0.04933	0.05122	0.05122	10	11.47
IoT-MotionLight	CNN	98.1	99.72	99.95	0.1149	0.0655	0.0656	100 (54)	54
	DNN	99.66	99.88	99.91	0.009	0.004	0.0025	100 (67)	67
	Tabnet	97.73	97.91	97.18	0.07678	0.0738	0.23802	10	181.18
IoT-garage_door	CNN	98.78	99.78	99.91	0.0763	0.0474	0.0437	100 (50)	50
	DNN	99.6	99.73	99.86	0.0102	0.005	0.0032	100 (37)	37
	Tabnet	88.835	89.554	90	0.06843	0.06645	0.06645	10	148.34
IoT-Modbus	CNN	98.95	100	100	0.0913	0.0603	0.0567	100 (32)	64
	DNN	100	100	100	7.75E-06	0	0	100(76)	76
	Tabnet	100	100	100	0.0015	0.0016	0.0016	5	50.79
IoT-GPS	CNN	99.46	100	100	0.054	0.0365	0.032	100 (47)	47
	DNN	99.62	99.92	100	0.011	0.0032	0.0012	100 (18)	19
	Tabnet	99.655	99.69	99.32	0.02952	0.02974	0.02974	13	181.61

**Table 2 sensors-25-00847-t002:** Performance evaluation of all models on the network dataset.

Model	Precision	Recall	F1 Score	Accuracy
CNN	98	98	98	99.24
DNN	98.18	98.19	98.17	99.03
TabNet	92	95	93	95

**Table 3 sensors-25-00847-t003:** Performance comparison of all three models on the IoT-Fridge dataset.

Model	Precision	Recall	F1 Score	Accuracy
CNN	99.907	99.92	99.92	99.91
DNN	100	100	100	100
TabNet	99.03	99.33	99.23	99.23

**Table 4 sensors-25-00847-t004:** Performance evaluation of all models for the IoT-Thermostat dataset.

Model	Precision	Recall	F1 Score	Accuracy
CNN	100	100	100	100
DNN	100	100	100	100
TabNet	98.95	97.45	98.05	98.21

**Table 5 sensors-25-00847-t005:** Performance comparison of all three models on the IoT-MotionLight dataset.

Model	Precision	Recall	F1 Score	Accuracy
CNN	99.96	99.96	99.96	99.95
DNN	99.93	99.93	99.93	99.91
TabNet	98.04	97.91	97.90	97.18

**Table 6 sensors-25-00847-t006:** Performance comparison of all three models on the IoT-garage_door dataset.

Model	Precision	Recall	F1 Score	Accuracy
CNN	99.92	99.93	99.93	99.91
DNN	99.90	99.90	99.90	99.86
TabNet	99.18	99.25	99.20	99

**Table 7 sensors-25-00847-t007:** Performance comparison of all three models on the IoT-Modbus dataset.

Model	Precision	Recall	F1 Score	Accuracy
CNN	100	100	100	100
DNN	100	100	100	100
TabNet	100	100	100	99.32

**Table 8 sensors-25-00847-t008:** Performance comparison of all three models on the IoT-GPS dataset.

Model	Precision	Recall	F1 Score	Accuracy
CNN	100	100	100	100
DNN	100	100	100	100
TabNet	99.53	99.17	99.33	99.32

**Table 9 sensors-25-00847-t009:** Proposed approach vs. benchmarked methods on TON_IoT dataset.

Authors	Method	Accuracy
Yaras and Dener [18]	CNN+LSTM	98.75% network
Li et al. [19]	Decision tree+feature selection	77.42% network
Krishnan and Shrinath [20]	Localized Generalized Matrix Learning Vector Quantization (LGMLVQ)	93.71% network
Awotunde et al. [21]	Stacking-ensemble model	99.09% network 99.73% highest IoT dataset
Alotaibi et al. [22]	Ensemble of ML	98.64%
Proposed	CNN	99.24 (network) 99.96% (average accuracy on IoT datasets)

## Data Availability

The original contributions presented in this study are included in the article. Further inquiries can be directed to the corresponding author(s).

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
