# Peer review of "An Intrusion Detection System over the IoT Data Streams Using eXplainable Artificial Intelligence (XAI)"

_sensors, 2025, doi:10.3390/s25030847_

Round 1

Reviewer 1 Report

Comments and Suggestions for Authors

The paper examines intrusion detection systems for IoT networks with the aim of comparing different deep learning methods for attack detection accuracy and feature explainability. The study selects three models to compare, two customized deep learning models and one non-customized transfer learning model from a previous study. Additionally, two explainable AI methods are selected for determining model features that are the main contributors to predictions made.

The motivations for the paper are well described and the literature review is sufficient. The experiments are clearly explained and well conducted. The following items are needed to improve the paper.

1) A discussion should be provided to elucidate why the three models (1D-CNN, DNN, and TabNet) were selected for comparison. Two of the models are customized; why is this necessary? Why were the particular customizations chosen to modify the 1D-CNN and DNN models?  Why was a transfer learning model selected for comparison with the deep learning models? Why was TabNet selected? Is TabNet the best transfer learning model to use for comparison? Why was the transfer learning model not also customized? Please provide further discussion that justifies the selections made.

2) Similarly, two explainable AI methods are selected for comparison of their ability to identify features that contribute to model predictions. Why are these two XAI methods selected? Do these represent the best methods to compare? Please provide further discussion that justifies the selections made.

Minor comments:

Figures 7 and 19 appear to have bad resolution. Table 5 appears to have a formatting problem.

Author Response

Respected Reviewer,

We would like to express our gratitude for your thorough analysis, which aided me in improving the paper's overall clarity and credibility. We are also grateful for the time and effort you put into this evaluation, which will be extremely beneficial to both me and the academic community. All the changes made in the revised manuscript are in red font.

Comments and Suggestions for Authors

The paper examines intrusion detection systems for IoT networks with the aim of comparing different deep learning methods for attack detection accuracy and feature explainability. The study selects three models to compare, two customized deep learning models and one non-customized transfer learning model from a previous study. Additionally, two explainable AI methods are selected for determining model features that are the main contributors to predictions made.

The motivations for the paper are well described and the literature review is sufficient. The experiments are clearly explained and well conducted. The following items are needed to improve the paper.

Comments 1:

  1. A discussion should be provided to elucidate why the three models (1D-CNN, DNN, and TabNet) were selected for comparison. Two of the models are customized; why is this necessary? Why were the particular customizations chosen to modify the 1D-CNN and DNN models? Why was a transfer learning model selected for comparison with the deep learning models? Why was TabNet selected? Is TabNet the best transfer learning model to use for comparison? Why was the transfer learning model not also customized? Please provide further discussion that justifies the selections made.

Response 1: Thanks for bringing the missing things to our notice.

The multiple experiments on different Machine learning and Deep Learning models were performed with different variations in model architecture.  But, the most promising results were for the 1D-CNN and DNN with the optimized architecture defined in the paper in Figures. 2 and 3, respectively. The hyperparameter tuning is performed to find the best hyperparameters for both the customized models.

The above paragraph is added at line 310 in the paper.

TabNet was selected to be compared with the Customized CNN and DNN models as the TabNet transfer learning model is a deep learning model designed for training on structured datasets. The TabNet is not customized, as the objective of the paper is to test the pre-trained model directly on the new dataset, i.e., the datasets used for the experiments, to verify its performance by fine-tuning the hyperparameters.

This paragraph is included in line 333 of the paper.

Comments 2:

2) Similarly, two explainable AI methods are selected to compare their ability to identify features that contribute to model predictions. Why are these two XAI chosen methods? Do these represent the best methods to compare? Please provide further discussion that justifies the selections made.

Response 2: 

LIME and SHAP are the most commonly used techniques for explaining the predictions in structured datasets. As all the datasets used for this research are structured, LIME and SHAP are selected and used.

This is added in the paper at line 352.

Comments 3:

  • Figures 7 and 19 appear to have bad resolution. Table 5 appears to have a formatting problem.

Response 3: Figures 7 and 9 are replaced with high-resolution images, and the formatting of Table 5 is checked and corrected.

Thank You.

Regards,

Adel AlAbbadi and Fuad Bajaber.

Reviewer 2 Report

Comments and Suggestions for Authors

Respected authors,

The main issue with the paper is consistency, as abbreviations, in-text references, and similar cases need to be checked in the entire paper. A pdf is attached with highlighted errors, mostly regarding the formatting of the paper and style issues. A problem that needs to be seriously addressed is the quality of images. Some are distorted, and some are just pixelated. Figures are not in order as after 7 goes 11, so please check this as well. All figures, tables, algorithms, and every other type of figure-like element you use must be properly referenced and elaborately described in the text. Some figures and tables are poorly designed liked the figures that are crossed out in the pdf and a better form of presenting this must be found. Every error highlighted in the pdf must be checked throughout the paper. The results seem interesting but ovefitting concerns are present. The authors only address this issue at one point when describing pooling layers. More attention must be paid to this and the authors should explain this better.

Kind regards,
Reviewer

Author Response

Respected Reviewer,

We would like to express our gratitude for your thorough analysis, which aided me in improving the paper's overall clarity and credibility. We are also grateful for the time and effort you put into this evaluation, which will be extremely beneficial to both me and the academic community. All the changes made in the revised manuscript are in red font.

Comments and Suggestions for Authors

The main issue with the paper is consistency, as abbreviations, in-text references, and similar cases need to be checked in the entire paper. A pdf is attached with highlighted errors, mostly regarding the formatting of the paper and style issues. A problem that needs to be seriously addressed is the quality of images. Some are distorted, and some are just pixelated. Figures are not in order as after 7 goes 11, so please check this as well. All figures, tables, algorithms, and every other type of figure-like element you use must be properly referenced and elaborately described in the text. Some figures and tables are poorly designed liked the figures that are crossed out in the pdf and a better form of presenting this must be found. Every error highlighted in the pdf must be checked throughout the paper. The results seem interesting but ovefitting concerns are present. The authors only address this issue at one point when describing pooling layers. More attention must be paid to this and the authors should explain this better.

Response to comments: I am really thankful to you for your keen attention to the details of the paper, which helps us improve the quality of images, tables, formatting, and the content of the paper and the presentation.

The formatting of the paper has been checked and corrected. Table 5 had an issue that was corrected in the revised version.

The revised version maintains the order of the figures, and all the figures with low resolution are replaced with good-resolution images. Special attention is given to the figures that are crossed in PDF. Figure 1 is changed as suggested.

All the abbreviations follow the lowercase, and consistency is maintained in the complete revised paper.

All the figures, tables, and algorithms are referred to and elaborated in the text.

Every error highlighted in the PDF is checked, and the paper is revised according to the suggestions. Each error is resolved and highlighted in red.

This paper tried to resolve the overfitting issues in the proposed models by utilizing batch normalization and dropout layers. Also, the dataset is pre-processed to achieve the highest performance. Still, the difference between the test accuracy and training accuracy for the network dataset is just 0.74% for CNN, and on the IoT dataset, it is 0.04% for CNN. So, the problem is not with the model’s architecture but with the dataset. As in the case of IoT datasets, 100% accuracy is achieved for four datasets, while for two datasets, i.e., IoT_motionlight, the accuracy achieved is 99.5, and for  IoT_garage door is 99.1 by the proposed CNN model.

The above paragraph is added in the last paragraph of the results and analysis section at line 636.

Thank You.

Regards,

Adel AlAbbadi and Fuad Bajaber.

Round 2

Reviewer 1 Report

Comments and Suggestions for Authors

Thanks to the authors for addressing the comments and improving the paper. 

Minor point:

In the discussion of Lime and Shape's selection as the XAI methods for comparison (line 352), it would be useful to add one or more references to studies or other sources that recognize their widespread use. This provides evidence that these XAI methods are prevalent and can be considered as "state-of-the-art". Alternatively you could list several studies that utilized either Lime or Shape for their work in XAI - this would also show that these methods are commonly used.

Author Response

Respected Reviewer,

We would like to express our gratitude for helping us find the missing details in the paper, which improved its overall clarity and credibility. All the changes made in the revised manuscript are in red font.

Comments and Suggestions for Authors

Minor point:

In the discussion of Lime and Shape's selection as the XAI methods for comparison (line 352), it would be useful to add one or more references to studies or other sources that recognize their widespread use. This provides evidence that these XAI methods are prevalent and can be considered as "state-of-the-art". Alternatively you could list several studies that utilized either Lime or Shape for their work in XAI - this would also show that these methods are commonly used.

Response to comment:  Thanks for highlighting it. The three references are added below.

  1. Gaspar, D.; Silva, P.; Silva, C. Explainable AI for Intrusion Detection Systems: LIME and SHAP Applicability on Multi-Layer Perceptron. IEEE Access 2024, 12, 30164–30175. https://doi.org/10.1109/access.2024.3368377.
  2. Salih, A.M.; Raisi-Estabragh, Z.; Boscolo Galazzo, I.; Radeva, P.; Petersen, S.E.; Menegaz, G.; Lekadir, K. A Perspective on Explainable Artificial Intelligence Methods: SHAP and LIME. Adv. Intell. Syst. 2024, 6, [Page Range if available]. https://doi.org/10.1002/aisy.202400304.
  3. Sharma, P.; Kumar, Y.; Varshney, H. Explainable AI for Intrusion Detection: Improving Model Transparency in Cybersecurity. IEEE Access 2020, 8, 181487–181502. https://doi.org/10.1109/ACCESS.2020.3029475.

Also, the first two lines at 352 are modified as given below.

LIME and SHAP are the most commonly used techniques for explaining the predictions in structured datasets, as suggested by Gaspar et al. [36] and Sharma et al. [37]. They are especially favorable methods for cybersecurity applications [38].

Thank You.

Regards,

Adel AlAbbadi and Fuad Bajaber.